# A model intercomparison of radiocarbon-based marine reservoir ages during the last 55 kyr including abrupt changes in the Atlantic Meridional Overturning Circulation

Peter Köhler<sup>1</sup>, Laurie Menviel<sup>2</sup>, Frerk Pöppelmeier<sup>3</sup>, Timothy J. Heaton<sup>4</sup>, Edouard Bard<sup>5</sup>, and Luke C. Skinner<sup>6</sup>

**Correspondence:** Peter Köhler (peter.koehler@awi.de)

Abstract. Changes in the marine reservoir age (MRA) of the surface ocean are important information used for radiocarbon dating of marine sediment cores or archaeological artifacts. MRA changes are expressed relative to the atmosphere, and as such are dependent on the prevailing atmospheric radiocarbon calibration curve. The most recent estimate for evolving global average MRA for latitudes approximately  $< 50^{\circ}$  is incorporated into the marine calibration curve Marine 20, which was directly calculated from the atmospheric  $\Delta^{14}$ C record, IntCal20, using the carbon cycle box model BICYCLE, taking into account observed changes in the carbon cycle. These simulations did not consider changes in the strength of the Atlantic meridional overturning circulation (AMOC) related to Dansgaard/Oeschger and Heinrich events. A recent study using the successor BICYCLE-SE suggested that abrupt AMOC changes would lead to changes in MRA of less than 100 <sup>14</sup>C yr in the non-polar surface ocean (about  $< 50^{\circ}$ ). To better support previous model-based MRA and to further constrain the impact of AMOC changes on MRA, we here assess transient simulations of the last 55 kyr performed by two Earth System Models of Intermediate Complexity (EMICs), LOVECLIM and Bern3D, and compare them to the published BICYCLE-SE box model results and previous output from the LSG ocean general circulation model. The setups within this MRA model intercomparison (MRA-MIP) are not identical, but all models are forced by atmospheric  $CO_2$  and  $\Delta^{14}C$  to have the surface ocean carbon cycle state as close as possible to reconstructions. Simulations with abrupt AMOC reductions during stadials display a rise in in MRA in the surface northern Atlantic (>50°N) and the deep Atlantic, for example during Heinrich stadial 1 of 300–1250 and 500–1300 <sup>14</sup>C yr, respectively, roughly in agreement with their reconstructed rises by about 1200-1300 <sup>14</sup>C yr. We find that the changes in the mean non-polar surface MRA ( $<50^{\circ}$  latitude) during abrupt AMOC changes in LOVECLIM are also in the order of  $\pm 100$ 

<sup>&</sup>lt;sup>1</sup>Alfred-Wegener-Institut Helmholtz-Zentrum für Polar-und Meeresforschung, P.O. Box 12 01 61, 27515 Bremerhaven, Germany

<sup>&</sup>lt;sup>2</sup>Climate Change Research Centre, Australian Centre for Excellence in Antarctic Science, University of New South Wales, Sydney, NSW, Australia

<sup>&</sup>lt;sup>3</sup>Climate and Environmental Physics, Physics Institute and Oeschger Centre for Climate Change Research, University of Bern, Bern, Switzerland

<sup>&</sup>lt;sup>4</sup>Department of Statistics, School of Mathematics, University of Leeds, Leeds, UK

<sup>&</sup>lt;sup>5</sup>CEREGE, Aix-Marseille University, CNRS, IRD, INRAE, Collège de France, Technopole de l'Arbois BP 80, Aix en Provence Cedex 4, France

<sup>&</sup>lt;sup>6</sup>Godwin Laboratory for Palaeoclimate Research, Earth Sciences Department, University of Cambridge, Downing Street, CB2 3EQ Cambridge, UK

 $^{14}$ C yr, while in Bern3D simulating changes are up to  $\pm 200^{-14}$ C yr. While the models tend to agree that a reduced AMOC leads to lower MRA in the low-latitude surface ocean, under some conditions the opposite is found (e.g. simulations with LOVECLIM across Heinrich stadial 1). Spatially resolved results of the models show that changes in surface MRA during stadials depict the general pattern of a radiocarbon bipolar seesaw (older surface water in the high north, younger in the high south and in the Indo-Pacific), in agreement with previously published reconstructions, but with model-specific details in the non-polar Atlantic. Throughout the last 50 kyr, the change in the multi-model mean in non-polar MRA of the 2 EMICs when compared with Marine20 is less than  $100^{-14}$ C years and within the uncertainties of Marine20. Furthermore, changes in the MRA of the high latitude Southern Ocean (>  $50^{\circ}$ S) are extremely model-dependent and for most times between 18 and 43 kyr BP the changes in the multi-model mean MRA are larger than the 95% confidence interval of the non-polar MRA depicted in Marine20, making the construction of a similarly numerical model-based calibration curve for this region a challenging task.

## 1 Introduction

Radiocarbon ( $^{14}$ C), with a half-life of about 5700 years, is an ideal tool for determining the age of carbonaceous materials and tracing components of the global carbon cycle over the last  $\sim$ 55,000 years (Hajdas et al., 2021). However, the amount of  $^{14}$ C in all reservoirs is not constant over time, but varies due to a changing carbon cycle and geomagnetic and solar effects on the  $^{14}$ C production rates in the upper atmosphere (e.g. Heaton et al., 2021; Köhler et al., 2022). Therefore, radiocarbon dating needs to rely on calibration curves, that take these temporal changes into consideration.

Within the last iteration of these calibration curves the carbon cycle box model BICYCLE (Köhler et al., 2006) has been used to calculate  $\Delta^{14}$ C in the ocean. For that effort the model has been forced by changes in atmospheric CO<sub>2</sub> (Figure 1a) as seen in ice cores (Köhler et al., 2017) and by changes in atmospheric  $\Delta^{14}$ C (Figure 1b) as compiled within IntCal20 (Reimer et al., 2020). From  $\Delta^{14}$ C in both the atmosphere and the surface ocean the marine reservoir age (MRA) of the non-polar surface ocean, also called the Marine 20 calibration curve (Heaton et al., 2020), has been constructed (Figure 1c), Here, BICYCLE was applied using a Monte-Carlo approach with 500 repetitions to account for uncertainties in data and the prescribed parametrization of piston velocity and the strength of the Atlantic meridional overturning circulation (AMOC), the two processes identified to be most important for simulated surface ocean  $\Delta^{14}$ C. So far, this approach ignored changes in the AMOC linked to the millennial-scale variability of Dansgaard/Oeschger and Heinrich events (Henry et al., 2016; Menviel et al., 2020), and thus climatic shifts observed in ice core records and marine sediment cores (e.g. Blunier and Brook, 2001; Davtian and Bard, 2023) (Figure 1d). However, considering abrupt AMOC changes in models can produce anomalies in radiocarbon age of the non-polar surface ocean of the order of  $\sim 100^{-14}$ C years (Köhler et al., 2024a). Although this is within the  $2\sigma$ uncertainty of Marine 20, reconstructions of past MRA variability demonstrate the occurrence of larger changes across the last deglaciation in association with millennial-scale climate anomalies (Skinner et al., 2019), which points to not yet fully considered errors in obtained simulations and larger uncertainties in marine radiocarbon calibrations around abrupt changes in AMOC strength.

50

60

To further explore the features and robustness of Marine20 — and its potential model-dependency — we here compare changes in surface ocean MRA, mainly in the non-polar areas (50°S to 50°N), over the last 55 kyr as simulated in a box model and two Earth system models of intermediate complexity (EMICs). We rely on initial results of BICYCLE-SE (Köhler et al., 2024a) and add outputs from more complex EMICs, as they are fast enough to transiently simulate the last 55 kyr in a reasonable amount of time. To this end we use the outputs from LOVECLIM (Menviel et al., 2014) and Bern3D (Pöppelmeier et al., 2023b). We also add some results of the available outputs from the LSG ocean general circulation model (OGCM) without abrupt AMOC changes, that have already been used within Marine20 (Butzin et al., 2020). These LSG runs are transiently forced with variable atmospheric carbon records ( $CO_2$ ,  $\Delta^{14}C$ ), but all else was kept constant. While the focus is on the non-polar areas, here defined as about  $

the plotted time series in the main text on changes in MRA with respect to pre-industrial (PI), since this is the target of the marine <sup>14</sup>C calibration curves, we show absolute values of MRA in time series in the SI, which might be more of an interest from a climate/carbon cycle perspective.

# 85 2.1 BICYCLE-SE

BICYCLE-SE is a carbon cycle box model consisting of 10 ocean boxes, a one box atmosphere and a 7 box terrestrial biosphere which also considers carbon exchange fluxes with the solid Earth by a sediment module, volcanic outgassing, weathering and coral reef growth. It is fully described in Köhler and Munhoven (2020) with the processes related to carbon isotopes being updated recently (Köhler and Mulitza, 2024). The simulation scenarios shown here have already been discussed in Köhler et al. (2024a). Scenario A0 is a run without abrupt changes in AMOC during Greenland stadials, while AMOC changes drastically in scenario A3. These AMOC changes are prescribed by the mean of two independent Iberian Margin sea surface temperature (SST) data sets (Davtian and Bard, 2023) which is then rescaled to obtain a minimal AMOC of 2 Sv during HS1, since this agrees best with <sup>14</sup>C reconstructions in the deep Atlantic ocean. See Figure S1 for details on both the contained climate change and the simulated MRA with BICYCLE-SE.

#### 95 **2.2 LOVECLIM**

100

110

LOVECLIM (Goosse et al., 2010) is an EMIC, which includes a quasi-geostrophic T21 atmospheric model, an ocean general circulation model ( $3^{\circ} \times 3^{\circ}$ , 20 vertical levels) coupled to a dynamic-thermodynamic sea-ice model, a vegetation model and a global carbon cycle model. The conditions at 54 kyr BP were obtained through a transient experiment starting at 140 kyr BP (Menviel et al., 2021). The model is forced with the transient evolution of orbital parameters (Berger, 1978), atmospheric greenhouse gas concentration (Köhler et al., 2017; Bereiter et al., 2015), and continental ice-sheet geometry and albedo. For the period 140 to 120 kyr BP, the evolution of the continental ice-sheet geometry and albedo are as described in the PMIP4 protocol of the penultimate deglaciation (Menviel et al., 2019). Between 120 kyr BP and 20 kyr BP, the continental ice-sheet geometry and albedo evolution are given by a 130 kyr BP off-line ice-sheet model simulation (Abe-Ouchi et al., 2007). Between 20 and 2 kyr BP, the model is forced by the continental ice-sheet geometry and albedo evolution from ICE4G (Peltier, 1994). At 54 kyr BP, the Bering Strait is closed, and is gradually opened between 11 and 10 kyr BP.

Between 140 and 54 kyr BP, the atmospheric  $\Delta^{14}C$  is set constant at 0‰. The model is then re-equilibrated with atmospheric  $\Delta^{14}C$  at 54 kyr BP for 5000 years. From 54 kyr BP the model is forced by the transient evolution of atmospheric  $\Delta^{14}C$  using the IntCal20 data (Reimer et al., 2020). To allow the ocean to fully equilibrate the oceanic  $\Delta^{14}C$  can be used in this context here from about 50 kyr BP onwards.

To simulate the millennial-scale variability of the last glacial period and the impact of deglacial ice-sheet disintegration (scenario fwf), meltwater of up to 0.3 Sv is added into the North Atlantic with the timing these events being based on Iberian Margin SST from Martrat et al. (2007). For comparison, a simulation without such meltwater fluxes and related millennial-scale changes is performed (scenario nofwf). See Figure S2 for details on both the contained climate change and the simulated MRA with LOVECLIM.

#### 115 2.3 Bern3D

The Bern3D model is a coarse resolution ( $68 \times 46 \times 40$  irregular spaced grid in longitude, latitude and depth) Earth system model (or EMIC) that couples a frictional-geostrophic ocean to 2D energy-moisture balance atmosphere. In contrast to the detailed description of the model in Pöppelmeier et al. (2023b), we here employ it without the dynamical ice-sheet component. Instead, ice-sheet evolution over the last 55 kyr is prescribed. For this, the Last Glacial Maximum (LGM) (ICE-6G (Peltier et al., 2015)) and modern ice-sheet extents are linearly interpolated based on the benthic  $\delta^{18}$ O record of Lisiecki and Raymo (2005) for the last 55 kyr. The biogeochemical cycle including the implementation of radiocarbon is described in Parekh et al. (2008) and Tschumi et al. (2008).

In addition, to changes in orbital configuration (Berger, 1978), greenhouse gases ( $CO_2$ ,  $CH_4$ , and  $N_2O$  (Köhler et al., 2017)), and ice-sheets, aerosol radiative forcing due to the changing atmospheric dust load is prescribed as well for the transient simulations. The temporal evolution of the aerosol radiative forcing follow the EPICA Dome C dust record (Lambert et al., 2012), which has been normalised to values between 0 during the late Holocene and -2 W m $^{-2}$ , which corresponds to an average radiative forcing of about -1 W m $^{-2}$  at the LGM (see Pöppelmeier et al., 2023b). In scenario PallSTD, additional freshwater fluxes are applied during all stadials, with a maximum freshwater flux of 0.4 Sv during Heinrich stadials (HS) and 0.2 Sv during non-Heinrich stadials. The timing of stadials is, similarly as in BICYCLE-SE, based on the Iberian Margin SST as published in Davtian and Bard (2023).

The model was first spun up to PI conditions, followed by a 10 kyr spinup to the conditions of 65 kyr BP. The model was then run transiently from 65 kyr BP to present day, with the time from 65 kyr BP to 55 kyr BP, serving as an additional transient spinup period (with atmospheric  $\Delta^{14}$ C fixed at its IntCal20 value for 55 kyr BP) to reach a realistic radiocarbon content in the deep ocean. The additional scenario Pnofwf is in all but the missing freshwater fluxes during stadials identical to PallSTD. The simulations shown here are due to the different spinup (and missing flux corrections) different to the run published in Pöppelmeier et al. (2023a), which has been compared to multi-proxies during Termination I. See Figure S3 for details on both the contained climate change and the simulated MRA with Bern3D.

# 2.4 LSG

140

Output from the Hamburg Large Scale Geostrophic OGCM (LSG-OGCM) has already been used within IntCal20 (Butzin et al., 2020; Heaton et al., 2020; Reimer et al., 2020). LSG has been used with a horizontal resolution of  $3.5^{\circ}$  and a vertical resolution of 22 unevenly spaced levels. Further setup details are contained in Butzin et al. (2005, 2020). No additional simulations (or outputs) have been performed. The model has been run with constant ocean circulation in three different climate setups for three different atmospheric  $\Delta^{14}$ C forcings. The three different climate setups contain a scenario that mimics present-day climate background conditions approximating the Holocene and interstadials. One glacial scenario aims at representing the LGM, features a shallower and by about 30% weaker AMOC compared to interglacial. A second glacial climate scenario mimics cold stadials with further AMOC weakening by about 60%. The three atmospheric  $\Delta^{14}$ C forcings contain the posterior mean estimate for the Hulu-based  $^{14}$ C atmospheric reconstruction and two time series that cover its 95% confidence interval

150

(CI) (mean  $\pm 2\sigma$ ). Here, the median MRA from these nine simulations is shown, which is also what has been used within IntCal20.

# 2.5 Marine <sup>14</sup>C data

For an initial evaluation of model-based surface MRA we compare simulations with data-based MRA from the Global Ocean Data Analysis Project (GLODAP) (Key et al., 2004), which is a synthesis of ocean sampling expeditions carried out in the 1990s within the World Ocean Circulation Experiment (WOCE), the Joint Global Ocean Flux Study (JGOFS), and the Ocean Atmosphere Carbon Exchange Study (OACES). Here, we calculate

155 MRA = 
$$8033 \cdot \ln \left( \frac{\frac{\Delta^{14} C_{\text{atmosphere}}}{1000} + 1}{\frac{\text{pre-bomb DI}^{14} C_{\text{abiotic}}}{\text{PI DICabiotic}}} \right)^{14} \text{C yr}$$
 (2)

based on the approach used for the ocean model intercomparison project (Orr et al., 2017). Abiotic PI dissolved inorganic carbon (DIC) has been derived from the atmospheric  $CO_2$  concentration of 284 ppm and pre-bomb abiotic dissolved inorganic  $^{14}C$  (DI $^{14}C$ ) from  $\Delta^{14}C_{atm} = -24\%$ . According to Graven et al. (2017), the latter represents the mean value of the decade before the onset of the bomb-peak in mid-1955, which has been used as pre-bomb baseline to reconstruct marine  $^{14}C$  from potential alkalinity (Rubin and Key, 2002). This so-called "natural" GLODAP-based MRA will always be a compromise since  $^{14}C$  is only corrected for bomb- $^{14}C$ , but not for fossil fuels, the so-called  $^{14}C$ -Suess effect (Suess, 1955; Stuiver and Quay, 1981).

To evaluate this natural GLODAP-based MRA we calculate local anomalies in MRA, the so-called  $\Delta R$  by subtracting the Marine20-based global MRA for 1950 CE (407  $^{14}$ C yr), and compare the resulting map of  $\Delta R$  with the entire content of the  $\Delta R$  data base found at http://calib.org/marine/ (Reimer and Reimer, 2001). We take the 2000 entries of this data base which were available with stated uncertainties on 2nd October 2025. The data were collected mostly between the years 1729 and 1959 CE with 3 entries from the 17th century, one from 1512 CE and one from before CE.  $\Delta R$  is calculated from the difference of the reverse-calibrated collection year using Marine20 and the measured  $^{14}$ C age and is assumed to stay constant in time. We average the data in a spatial resolution of  $2^{\circ}$  in both latitude and longitude ending with 609 values. Here, we use (as in the application of the online scripts of the data base) weighted means for averaging with the mean of  $\Delta R$  ( $\overline{\Delta R}$ ) given by

$$\overline{\Delta R} = \frac{\sum_{i} \frac{\Delta R_{i}}{\sigma_{i}^{2}}}{\sum_{i} \frac{1}{\sigma_{i}^{2}}}$$
(3)

where  $\sigma_i$  is the uncertainty in  $\Delta R_i$  and the reported uncertainty  $\sigma$  of  $\overline{\Delta R}$  is the maximum of the standard deviation of  $\overline{\Delta R}$  and the weighted uncertainty in  $\overline{\Delta R}$  (see http://calib.org/marine/AverageDeltaR.html or Bevington (1969) for details). We plot not only  $\overline{\Delta R}$ , but also a minimum and a maximum version with  $\overline{\Delta R} \pm 1\sigma$  (Figure S4). Independent of which version of the data-based  $\Delta R$  we take we find in general a good agreement between them and natural GLODAP (differences of typically up to  $100^{-14}$ C years) with the exception of single data points and the entire the west coast of North America, where values in the data base are more than  $100^{-14}$ C years older than in GLODAP, potentially caused by coastal effects.

185

200

In order to compare model outputs with observations, we make use of compiled deglacial marine radiocarbon data from Skinner et al. (2023) and regional time-series splines from Skinner et al. (2019). The latter include regional time series for the surface northeast Atlantic (>52°N and east of 24°W) and the Iberian Margin (~38°N, 10°W). We use the deep Atlantic MRA (i.e. B-Atm, deeper than 2 km water depth) based on the baseline selection of Skinner et al. (2023) in the realisation of Köhler et al. (2024a). We do not compare model results to Stern and Lisiecki (2013), since they used age models based on IntCal09 and the resulting 42 kyr-long time series of surface MRA cover 0–65°N in the Atlantic, a value which cannot be extracted from the box model simulations.

Additionally, point-wise surface MRA for the time slices of the LGM (19–21.8 kyr BP) and the HS1 (15–17.5 kyr BP) from the baseline selection of the data base of Skinner et al. (2023) are used for further model evaluation. Here, multiple MRA entries for the same sites are averaged using weighted means (similar to Eq. 3) reducing 67 entries for the LGM to 19 values (89 entries for the HS1 to 22 values), from which 13 exist for the same location in both periods making the calculation of HS1–LGM differences in MRA possible. Here, one entry for LGM was rejected as outlier since its MRA was > 1000 <sup>14</sup>C yr offset from four alternative MRAs for the same site. Most surface MRA data are from the northeast Atlantic (between Iberian Margin and Iceland) which are complemented with data from single sites in the Southern Ocean, the tropical East- and West Pacific and the tropical Atlantic.

# 3 Results and Discussions

## 3.1 Pre-industrial MRA compared to GLODAP

Estimates of changes in surface MRA are necessary for <sup>14</sup>C dating of marine carbon-containing material. For radiocarbon dating of marine organisms, such as foraminifera, it is important to take into account the organism's seasonal/depth habitat, e.g. for the estimation of appropriate local variations in reservoir age, that is  $\Delta R$  (Reimer and Reimer, 2001), or for comparison with model-based MRA estimates. Planktonic foraminifer habitats are species-specific, and often related to the mixed layer depth, but sometimes extend to greater depths (Kimoto, 2015). The mixed layer depth has a seasonal cycle, but for modern day latitudes  $< 50^{\circ}$ , it is typically less than 200 m (de Boyer-Montégut et al., 2004), which in our two EMICs is indeed the case for the annual mean mixed layer depths during the PI (Figure S5). Note, that due to a lack of more recent results from LOVECLIM, results for 2 kyr BP are chosen to stand in for the PI reference. The mean surface MRA calculated for the top 50 m within IntCal20 from LSG output (Butzin et al., 2020) might be relevant to foraminifer data from the lower end of the depth range, while the 100 m thick non-polar surface ocean boxes of the BICYCLE model used within Marine20 were probably in the middle of the relevant depth range. From the GLODAP data we calculate a mean surface MRA for latitudes  $< 50^{\circ}$  of 291, 318 and 356 <sup>14</sup>C yr, when considering data from the top 50, 100, and 200 m water depth, respectively, illustrating the centennialscale of the uncertainty related to the assumed habitat depth range. Furthermore, calculated surface MRA for different depth ranges from the transient simulations provide insights into how this depth-dependency might vary with time (Figures S2–S3). Here, the depth ranges over which surface MRA have been calculated differ in detail (Table 1), depending on model grid, since we only consider full vertical layers. Differences between MRA based on roughly the top 50 m or 100 m are similar, and in

agreement with GLODAP, while those based on roughly the top 200 m are about 50 and 100  $^{14}$ C yr higher than MRA of top 50 m in Bern3D and LOVECLIM, respectively.

Maps of surface MRA for 2 kyr BP (our PI reference) compare well with MRA based on natural  $^{14}$ C in GLODAP (Figure 2). Note, that atmospheric  $\Delta^{14}$ C at 2 kyr BP was with -16% only slightly different from the pre-bomb value of -25% in 1950 CE (0 kyr BP). However, be aware that this comparison has certain weaknesses due to the compromises in the GLODAP data (not free of the  $^{14}$ C-Suess effect, see Methods section for details) and our choice of using simulation results at 2 kyr BP. For latitudes  $

255

260

265

270

275

approach to about 11 Sv (Pöppelmeier et al., 2023a) is now  $\sim$ 15 Sv, since flux corrections applied in the other study have been neglected here.

The simulated non-polar MRA ( $<50^{\circ}$ ) contains remarkable model-specific differences. From the absolute values we can identify the model-specific offsets with Bern3D simulating the lowest PI MRA, while BICYCLE-SE simulates the highest PI MRA (Figure 3d). On the other hand, at the LGM (20 kyr BP), the smallest MRA is simulated by LSG, followed by Bern3D, LOVECLIM, and BICYCLE-SE.

Across the last glacial period, Bern3D simulates the largest changes in MRA, with a 1000 <sup>14</sup>C yr anomaly with respect to PI during the Laschamps geomagnetic excursion around 42 kyr BP (Simon et al., 2020), while LOVECLIM and LSG display the smallest MRA anomalies with respect to PI (Figure 3e). The differences to Marine20 — when plotted with respect to PI — show no consistent pattern, but to a large degree model-specific responses which nearly all fall in their sizes within the 95% CI of Marine20 (Figure 3f).

Only when we calculate differences from simulations with and without abrupt AMOC shutdown during stadials for the same model we find the tendency of smaller non-polar surface MRA during stadials with the notable exception of HS1 in LOVECLIM showing a more complex (rather opposite) dynamic (Figure 3g). Bard (1988) showed that  $\Delta$ MRA remains small ( $\sim 100^{-14}$ C yr) when changing the eddy diffusivity in the box-diffusion model (Oeschger et al., 1975) that has been widely used to convert atmospheric  $\Delta^{14}$ C changes in terms of oceanic changes (Stuiver et al. (1986) and subsequent IntCal calibration iteration until 2013). The main point outlined by Bard (1988) is that the global average surface MRA and mean deep ocean  $^{14}$ C age vary in opposite directions when eddy diffusivity changes, mimicking global overturning variations (all else being equal). Interestingly, scaling the eddy diffusivity to our AMOC proxy curve (SST record by Davtian and Bard, 2023) leads to a similar  $\Delta$ MRA pattern in the outcrop-diffusion box model as in the other models when they are averaged in the low-latitude regime (Figure 3g).

When plotting changes in MRA reconstructions for the LGM (Skinner et al., 2023) together with model outputs we find that most models underestimate for most locations the changes in surface MRA (Figure 4). The best agreement, especially in the high latitude, is obtained for LSG. This comparison is rather limited, due to the availability of only 19 data points. It has as additional caveat that simulations are focused on 20 kyr BP, while the data cover the wider time window of 19–21.8 kyr BP. Changes in models and data only roughly agree in a few areas (Iberian Margin, Caribbean) but elsewhere differ widely, with models underestimating the MRA derived from proxy records. Note that the about 10 data point in the northeast Atlantic show partly very different changes, ranging from a reduction in MRA to a rise by more than 1000 <sup>14</sup>C yr. This local diversity of the reconstructions challenges our model-data comparison since small-scale localised effects seen in the data might not be contained in the coarsely resolved models. Notably, there is also significant disagreement between models in particular regions. One reason for the data-model offsets might be the habitat depth of the planktonic foraminifera, even though the EMICs give surface MRA that differ by less than 50 <sup>14</sup>C yr when based on roughly the top 200 m instead of the top 50 m used so far (Figure S6). Interestingly, annual mean mixed layer depth at 20 kyr BP in the two EMICs differ in the non-polar regime only very little from the PI control (Figure S5). Another reason may be that the models are all missing key processes that might

280

305

influence vertical diffusivity in the upper ocean or air-sea gas exchange at the surface, particularly at high latitudes, or that they tend to simulate a different radiocarbon distribution in the interior ocean than prevailed in the past.

## 3.3 Details of the impacts of AMOC weakening on MRA

To further elucidate the model-specific responses for HS1, we compare simulations with reconstructions for that time interval (Figure 5). Here, we focus on the HS1-LGM difference. For this comparison, 13 data points exist in the data base of Skinner et al. (2023). Both EMICs show a general MRA decrease at mid and low latitudes notably in the Indo-Pacific. This almost global MRA reduction is compatible with that obtained with the BICYCLE-SE model. In addition to this common pattern, the models exhibit dynamical behaviour that is responsible for an increase of MRA in the northern North Atlantic, with varying intensity and spatial extent: confined to the northern North Atlantic (+Arctic) in BICYCLE-SE, intermediate in Bern3D, and widespread to most of the Atlantic basin in LOVECLIM. Thus, the different response in the Atlantic in HS1 readily explains LOVECLIMs different dynamics when comparing non-polar averages of runs with and without abrupt AMOC shutdown (Figure 3g). There are areas where models disagree with available data. In particular, the Iberian margin data, e.g. shown previously in Skinner et al. (2014, 2019, 2021), exhibit a clear MRA increase during HS1, as already acknowledged in Heaton et al. (2020), which is covered in LOVECLIM, but not in Bern3D. Both EMICs show the existence of a radiocarbon bipolar seesaw pattern, with widespread decrease in the Southern Ocean, not restricted to the Atlantic sector. This radiocarbon bipolar seesaw pattern with MRA increases in the northern Atlantic, which is connected with reduced MLD there (Figure S5, right column), and MRA reductions in the Southern Ocean is very reminiscent of the thermal bipolar seesaw (Stocker and Johnsen, 2003) available from paleo-SST data (e.g. Barker et al., 2009; Davtian and Bard, 2023) and from numerous model experiments (e.g. Zhang et al., 2014; Pedro et al., 2022). Indeed, this seesaw pattern in MRA variability is directly consistent with what has been previously observed in direct MRA reconstructions (Skinner et al., 2014, 2019, 2021, 2023; Skinner and Bard, 2022), where it has been hypothesized to relate to sea-ice variability in each hemisphere.

For some areas where the models disagree reconstruction-based evidence exist, especially from the Iberian Margin. However, for most of the equatorial Atlantic, where both models show the opposite change in MRA a data-based evaluation of the simulations is due to a lack of data still missing.

An alternative to calculating the MRA anomaly during HS1 is based on the different surface MRA at 16 kyr BP from simulations with and without abrupt AMOC shutdowns (Figure S7, right column). However, such a model-based anomaly cannot directly be compared with reconstructions. Here, results from all contributing models (BICYCLE-SE, LOVECLIM, Bern3D) show more positive  $\Delta$ MRA than in the calculations based on the HS1–LGM difference, but contain similar patterns. Increases in surface MRA in the non-polar Atlantic basin are now also contained in BICYCLE-SE.

As a final step, we compare simulated and reconstructed time series of surface northern Atlantic (>50°N) and deep Atlantic MRA — all with respect to PI (Figure 6). The maximum lengths of reconstructed timeseries are limited to 25–30 kyr, but they provide useful insights for Termination I, including HS1. As expected, simulations with abrupt AMOC changes contain higher  $\Delta$ MRA during HS1 than those without these abrupt changes. However, only LOVECLIM reached changes of 1300  $^{14}$ C yr in the surface northern Atlantic as found in the reconstructions, and only Bern3D reaches the HS1 peak of 1200  $^{14}$ C yr in the

315

320

deep Atlantic, although some thousand years later. The model-data offset at the surface northern Atlantic might be contributed to aliasing of spatial patterns, whereby the compiled data are largely restricted to east of 24°W in the northern Atlantic >42°N whereas model results include the whole northern Atlantic and Arctic Ocean (>50°N). However, if this was the case, MRA variability in the western North Atlantic (and Arctic) would need to have been greatly reduced compared to the eastern part of the basin, which is opposite to what the model simulations tend to suggest (Figure 5). Note that in BICYCLE-SE,  $\Delta$ MRA in both surface northern Atlantic and deep Atlantic are very similar probably linked to the box model geometry and prescribed fluxes which contain a direct connection and vigorous exchanges between both boxes. All models tend to agree on having peaks in both surface northern Atlantic and deep Atlantic  $\Delta$ MRA connected with millennial-scale climate change. However, amplitudes are model-specific with Bern3D simulating the largest amplitudes which even appear, although in smaller size, in the simulation without abrupt AMOC changes. This indicates (as already notified in Köhler et al., 2024a) that they are partly related to the variability contained in either atmospheric  $\Delta$ 14C or CO<sub>2</sub> and not only to abrupt AMOC changes, though the impact of such atmospheric effects is likely limited to less than  $\sim$ 300  $^{14}$ C years equivalent for the global deep ocean and far less for the surface ocean (Skinner et al., 2023). Possible explanations might be related to changes in the terrestrial carbon cycle (e.g. Menking et al., 2022; Wu et al., 2022).

# 3.4 Towards global MRA

Finally, we calculate changes in surface MRA from a multi-model mean (MMM) based on low latitude (<50°) results of scenarios with abrupt AMOC changes in the two EMICs, which are then compared with Marine20 (Figure 7a,b). While the full range of simulation results is nearly always within the 95% CI of Marine20, it is on its lower edge since the LGM. Furthermore, the MMM is consistently about 100 <sup>14</sup>C yr smaller than Marine20. We tested if this offset might be based on the chosen depth of the analysed surface MRA. However, we only found a slightly better agreement between MMM and Marine20 when using results from the mean of the top 200 m instead of the top 50 m as done in the default case (Figure 7a). This offset between MMM and Marine20 again illustrates that the pre-bomb non-polar MRA, which was about 407 <sup>14</sup>C yr for the year 1950 CE when using BICYCLE within Marine20 (Heaton et al., 2020) is model-specific.

In a very last step we additionally calculate changes in the surface MRA of the high latitudes (>50°) and of the global mean in order to understand how much they differ from the MRA of the non-polar areas (Figure S8). The MRA in both high northern and high southern latitudes in LSG contain a jump by more than 1000 <sup>14</sup>C yr at 10.7 kyr BP; however, this jump is related to an arbitrary change to an 'interglacial' overturning scenario from the reduced 'glacial' overturning scenario in the LSG model. As the much larger MRA at the LGM in high latitudes in LSG agreed better with reconstructions than the smaller values of the EMICs (Figure 4), this raises some questions: if these sparse LGM-based reconstructions are broadly representative of glacial conditions, one might ask if the EMICs are missing or inadequately resolving key processes in the high latitudes during glacial times. Alternatively, if the EMIC-based glacial polar MRA turn out to be more accurate than the LSG-based results any polar age calibration based on the latter (Heaton et al., 2023) are then potentially biased towards too old values. For the two EMICs, the global mean MRA and the non-polar MRA are very similar (Figure S8) suggesting that the non-polar MRA might indeed be of global applicability. However, when looking to the details we find that the range and the MMM of the changes in the

northern polar MRA (which is similar, but not identical to the MRA of the northern North Atlantic discussed earlier) is — apart from certain HSs with several centuries older MRA — indeed comparable with the 95% CI of Marine20 (Figure 7c). The changes in the southern polar MRA are much higher than in Marine20, both in range and in MMM (Figure 7d), especially in Bern3D although still by a few centuries younger than in LSG (Figure S8). Thus, a global usage of the non-polar MRA would lead especially in the Southern Ocean (but also at the Iberian Margin; Skinner et al., 2019) to underestimations of MRA by at least one century leading to an overestimate of calendar age in probes whose radiocarbon age calibration is based on them. This Southern Ocean age offset in the models is probably related to the higher glacial sea ice coverage (Figure S9) and the larger glacial MLD (Figure S5). Here, larger glacial summer sea ice in the Southern Ocean and larger glacial MLD in Bern3D compared to LOVECLIM potentially explain at least part of the model-specific ΔMRA in the Southern Ocean.

#### 4 Conclusions

Comparing for the first time results from transient simulations of two EMICs, 1 OGCM and 1 box model over the radiocarbon time window we find that abrupt AMOC changes might introduce anomalies of up to  $100\text{-}200^{-14}\text{C}$  yr to the changes in the mean non-polar surface MRA. Our results show, that especially the surface  $\Delta$ MRA in the non-polar Atlantic is model-specific and potentially different from the Indo-Pacific. Although existing MRA data for evaluating model behaviour are relatively sparse for Heinrich stadials (here, especially HS1) (e.g. Skinner et al., 2019, 2023), they suggest that the simulated  $\Delta$ MRA during HS1 in LOVECLIM appear to be closer to the reconstructions than in Bern3D. The model-intercomparison shows where different models have strength and weaknesses, but leaves still open, which might be the best approach in the next iteration of IntCal, e.g. calculating multi-model means or relying on individual models. The very different changes in glacial polar MRA (especially in the Southern Ocean) in the two EMICs and LSG indicates that our ability to simulate radiocarbon in the polar regions remains limited and a robust conclusion on their MRA changes remains dependent on a relatively sparse observational database.

For marine radiocarbon calibrations temporal changes in global average surface MRA (e.g. as provided by models such as in this study) may be combined with regional  $\Delta R$  estimates (for details see Heaton et al., 2020). However, this method requires that the regional  $\Delta R$ s remain constant over time, whereas a growing body of observations suggests that this is not the case over the last deglaciation and during HS1 in particular (e.g. Siani et al., 2013; de la Fuente et al., 2015; Skinner et al., 2014, 2019, 2023). Our mini-MRA-MIP supports this observation (as yet based on relatively sparse data), and suggests that during Greenland stadials MRA in the Atlantic varies differently than in the Indo-Pacific. While this points to the potential usefulness of regional calibration curves (Skinner et al., 2019; Mârza et al., 2024), an alternative and more immediately practical approach might be to continue to use one global calibration curve (as done so far) and apply larger and appropriately structured age uncertainties for different regions and time periods, such as the northern Atlantic during Greenland stadials.

Data availability. All simulation results are available from PANGAEA. BICYCLE(-SE), LSG and marine data splines: (Butzin et al., 2019; Heaton et al., 2020; Köhler et al., 2024b). Bern3D and LOVECLIM: registration with PANGAEA pending, temporary link to data (1 tgz file with < 1 MB): https://my.hidrive.com/lnk/EhfGOFikK.

Author contributions. PK designed the study, performed the BICYCLE-SE simulations, made all figures and led the writing of the draft.
LM performed LOVECLIM simulations. FP performed Bern3D simulations. EB performed box diffusion model simulations and contributed knowledge on SST data, marine <sup>14</sup>C data and MRA. LCS contributed knowledge on marine <sup>14</sup>C data and MRA. TH contributed insights how the findings might be used within the next iteration of IntCal. All authors discussed the results.

Competing interests. LM and LCS are members of the editorial board of Climate of the Past. Apart from that, the authors declare no competing interests.

Acknowledgements. We thank Martin Butzin for digged out details on natural <sup>14</sup>C within GLODAP and for ongoing discussions on <sup>14</sup>C and on LSG output and comments on the draft. We thank Florian Adolphi for comments on an earlier version of the draft. LM acknowledges funding from the Australian Research Council (ARC) grant SR200100008. EB is funded by the ANR project MARCARA. FP is funded by the European Union's Horizon Europe research and innovation program under Grant Agreement 101137601 (ClimTip) and 101184070 (Past-To-Future). LCS is acknowledges funding and support from the Royal Society and NERC grant NE/V011464/1.

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

**Table 1.** Overview on used models and simulation scenarios. The control (CTRL) scenarios have no abrupt AMOC changes, but AMOC changes rapidly in "abrupt" scenarios. The "top layers" indicate for which top surface water depth the MRA have been calculated. The target was 50, 100 and 200 m, but results differ due to different model grids.

| Model      | Туре      | Scenario |         | Used Top Layers and Comment                     | Citation              |
|------------|-----------|----------|---------|-------------------------------------------------|-----------------------|
|            |           | CTRL     | abrupt  |                                                 |                       |
| BICYCLE-SE | box model | A0       | A3      | 100 m (non-polar); 1000 m (polar)               | Köhler et al. (2024a) |
| LOVECLIM   | EMIC      | nofwf    | fwf     | 54 m, 104 m, 252 m                              | this study            |
| Bern3D     | EMIC      | Pnofwf   | PallSTD | 64 m, 99 m, 216 m                               | this study            |
| LSG        | OGCM      | med      | -       | 50 m, median of nine simulations without abrupt | Butzin et al. (2020)  |
|            |           |          |         | AMOC changes                                    |                       |

Figure 1. Relevant time series across the last 55 kyr. (a) Atmospheric  $CO_2$  spline based on multi-records (Köhler et al., 2017) as used by the models. (b) Atmospheric  $\Delta^{14}C$  from IntCal20 (Reimer et al., 2020) and (red points) an added extension from new kauri-based data around 42 cal kBP (Cooper et al., 2021). (c) Non-polar marine reservoir age (MRA) Marine20 (Heaton et al., 2020). For  $CO_2$ , IntCal20 and Marine20 the mean values and the 95% CI are plotted. Vertical bands mark Heinrich events (blue) or non-Heinrich stadials (pink) defined by (d) Iberian Margin SST (mean of UK37' and RI-OH' SST records) (Davtian and Bard, 2023). Heinrich events and the Younger Dryas (YD) and Greenland stadials (GS, Rasmussen et al. (2014)) with their numbers are labelled on the top ignoring GS 2.1b and 2.1c which fall into the LGM.

**Figure 2.** Surface MRA (<sup>14</sup>C yr) from all models (left) at 2 kyr BP for runs with abrupt AMOC changes (scenarios: A3@BICYCLE-SE, fwf@LOVECLIM; PallSTD@Bern3D) and med@LSG; (right) differences in surface MRA from natural GLODAP (top middle). BICYCLE-SE: surface boxes; all else: mean values of roughly the top 50 m. Use left color-code (brownish) for absolute values and right color-code (blue-to-red) for differences.

Figure 3. Combining model outputs with focus on non-polar surface ocean (see SI for time series sorted by model). Changes in (a) global mean sea surface temperature (GMSST) with respect to (wrt) to the most recent points of the time series and (b) AMOC strength in the investigated model runs. (c) Atmospheric  $\Delta^{14}$ C from IntCal20 (Reimer et al., 2020) for comparison. (d) MRA of the non-polar surface ocean (surface box or roughly the top 50 m) ranging from about 50°S to 50°N for the different simulations and Marine20 (Heaton et al., 2020). Differences in simulated non-polar surface MRA (e) wrt 2 kyr BP (PI) and (f) to Marine20 wrt 2 kyr BP. Grey background is the 95% CI of Marine20. (g) Differences in non-polar surface MRA between simulations with and without abrupt AMOC changes for BICYCLE-SE, LOVECLIM, Bern3D. Additionally,  $\Delta$ MRA from a box diffusion model (Bard, 1988) in which eddy diffusity is linearly related to our AMOC proxy (Iberian Margin SST of Davtian and Bard, 2023). Annual output from LOVECLIM for GMSST and AMOC is plotted as 50-yr running mean, output of Bern3D comes in timesteps of 50 yr. Vertical bands mark Heinrich events (blue) or non-Heinrich stadials (pink), see caption to Figure 1 for details. Results from simulations without abrupt AMOC changes have been omitted in (d–f) for clarity, but are found in the SI (Figures S1–S3).

**Figure 4.** Difference in surface MRA (<sup>14</sup>C yr) in all models (BICYCLE-SE: surface box; all else: roughly the top 50 m) between 20 kyr BP and 02 kyr BP (20 – 02 kyr BP) for runs with abrupt AMOC changes (scenarios: A3@BICYCLE-SE, fwf@LOVECLIM; PallSTD@Bern3D) and med@LSG. The 19 points in each panel are the difference for the LGM (19–21.8 kyr BP) based on reconstructions (Skinner et al., 2023) from natural GLODAP.

**Figure 5.** Differences in surface MRA (<sup>14</sup>C yr) from all models but LSG (BICYCLE-SE: surface box; all else: roughly the top 50 m) between 16 kyr BP and 20 kyr BP (16 – 20 kyr BP) for runs with abrupt AMOC changes (scenarios: A3@BICYCLE-SE, fwf@LOVECLIM; PallSTD@Bern3D). The 13 points in each panel are differences in MRA for HS1 (15–17.5 kyr BP) – LGM (19–21.8 kyr BP) based on reconstructions (Skinner et al., 2023).

**Figure 6.** MRA changes with respect to the most recent points of the time series in simulated (a) surface northern (>50° N) and deep (b) Atlantic. Combining all model outputs with focus on the Atlantic. The surface North Atlantic MRA (surface box or roughly the top 50 m) covers all north of 50°N including the Arctic Ocean. The deep Atlantic contains water below 2 km water depth. Data source: surface North Atlantic MRA: Skinner et al. (2019); deep Atlantic: Skinner et al. (2023) as compiled in Köhler et al. (2024a). Vertical bands mark Heinrich events (blue) or non-Heinrich stadials (pink), see caption to Figure 1 for details.

Figure 7. (a) Multi-model mean (MMM) of non-polar surface MRA from the 2 EMICs with simulations containing abrupt AMOC changes (fwf@LOVECLIM; PallSTD@Bern3D) in comparison to Marine20. Difference of MMM with respect to (wrt) 2 kyr BP (PI) of (b) non-polar surface MRA and of (c) northern and (d) southern polar surface ocean to changes in MRA of the mean of Marine20 also wrt 2 kyr BP. Light coloured areas show 95% CI for Marine20 and the full range of model results for the MMM. MMM is based on MRA results of roughly the top 50 m of the water column. Subfigure (a) contains also a version of the MMM based on roughly the top ~200 m. The MMM is only shown between 50 kyr BP and 2 kyr BP, since before 50 kyr BP spin-up effects take places, and at 2 kyr BP the LOVECLIM simulation ends. Vertical bands mark Heinrich events (blue) or non-Heinrich stadials (pink), see caption to Figure 1 for details.