# Peer review of "A model intercomparison of radiocarbon-based marine reservoir ages during the last 55 kyr including abrupt changes in the Atlantic Meridional Overturning Circulation"

_EGUsphere, 2025_

## Author Comment (AC1)

**Authors Replies to RC1**: ['Comment on egusphere-2025-5136'](), Patrick Rafter, 13 Dec 2025
Today: 13 Jan 2026

We copy the comments of RC1 Patrick Rafter below and add our authors replies (AR).

Summary
Although I'm not a modeling expert, but I am a user of these datasets and this MIP manuscript is timely and well-written. Given all that I've learned in this manuscript (e.g., the "surface ocean" is not consistent between the models), the observation that these models are broadly showing similar values (at least for the lower latitudes) is pretty amazing. This MIP is mostly focused on large-scale observations, with a mostly qualitative discussion of the inter-model differences and/or comparisons with observations. But maybe that's fine for this first of its kind work. Otherwise, I'm happy to see this useful study published.

AR: We thank Patrick Rafter for his effort in reviewing our study and for this overall positive evaluation of the draft.
We agree that the overall consistency between the models is impressive - and supports the idea that:
- We have captured many important physical things in these various models.
- The endeavour of a marine calibration curve is worthwhile for the community and that our work is improving its reliability.
We will add a sentence summarizing these details in the conclusion.

**Line by line notes:**

Line 12: Should define LSG earlier in the manuscript

AR: Although line 12 is still in the abstract the acronym LSG for the Large Scale Geostrophic Ocean General Circulation Model will be explained here as requested by the reviewer.

19: "leads to a lower MRA" is too vague; should be supported with values

AR: We will change this sentence to "... the models tend to agree that a reduced AMOC leads to lower MRA of about 100–300 $^{14}$C yr in the low-latitude surface ocean...".

44: Incomplete sentence here

AR: The sentence ending in line 44 is: "So far, this approach ignored changes in the AMOC linked to the millennial-scale variability of Dansgaard/Oeschger and Heinrich events (Henry et al., 2016; Menviel et al., 2020), and thus climatic shifts observed in ice core records and marine sediment cores (e.g. Blunier and Brook, 2001; Davtian and Bard, 2023) (Figure 1d)."
For a better understanding we will add words in the second half and will change it into: "So far, this approach ignored changes in the AMOC linked to the millennial-scale variability of Dansgaard/Oeschger and Heinrich events (Henry et al., 2016; Menviel et al., 2020), and thus climatic shifts observed in ice core records and marine sediment cores (e.g. Blunier and Brook, 2001; Davtian and Bard, 2023) (Figure 1d) were not included."

55: I believe that "LSG" is still undefined at this point in the manuscript!

AR: LSG will be defined in the abstract now (line 12), and its definition will be repeated here (and not anymore in section 2.4, where it was defined so far).

62: I like "come-as-you-are"

AR: Thanks!

70: The formatting of this equation makes it seem like it is all multiplied by "14C yr" when that is, in fact, the units of the equation. I see this in the later equation as well.

AR: Having the units in the equation is normally the best way to avoid confusion, but somehow it did not work here. Thus, we will shift the information on the units to the sentences prior to or after the equations 1 and 2.

155: yes, once again this formatting could be confusing

AR: Will be shifted, see previous reply.

178: I think it's generally understood this publication is using planktic foraminifera radiocarbon compilation, so I think it would help to be explicit about this?

AR: This sentence reads "In order to compare model outputs with observations, we make use of compiled deglacial marine radiocarbon data from Skinner et al. (2023) and regional time-series splines from Skinner et al. (2019)." While the regional time-series splines from Skinner et al. (2019) are indeed only from planktic foraminifera, the radiocarbon data from Skinner et al. (2023) are from planktic foraminifera for surface ocean MRA, but they also include data derived from benthic foraminifera for changes in MRA of the deep Atlantic. We will therefore change this sentence into "In order to compare model outputs with observations, we make use of compiled deglacial marine radiocarbon data from Skinner et al. (2023) and regional time-series splines based on planktic foraminifera from Skinner et al. (2019)."

193: Should just be "Results and Discussion"?

AR: Ok (was "Results and Discussions" so far).

213: Statistics could be used instead of "compare well" here

AR: This comment refers to "Maps of surface MRA for 2 kyr BP (our PI reference) compare well with MRA based on natural $^{14}$C in GLODAP (Figure 2)." In addition to the plotted surface MRA difference between simulated PI reference and data-based natural $^{14}$C, which can then be discussed in detail in the following paragraph (as done here), we will add the

area-weighted root mean square errors of the residuals of the model-based differences to natural $^{14}$C in GLODAP (202, 133, 86 and 265 $^{14}$C yr for BICYCLE-SE, LOVECLIM, Bern3D and LSG, respectively). These residual RMSEs are all substantially smaller than the 399 $^{14}$C yr of the area-weighted root mean square of the natural $^{14}$C in GLODAP evidencing the explanatory power of all the four models.

246: I think this run of one sentence paragraphs could be streamlined.

AR: We will merge the two paragraphs in lines 246-254 into one paragraph.

255: A run-on sentence, I believe.

AR: This comment refers to "Only when we calculate differences from simulations with and without abrupt AMOC shutdown during stadials for the same model we find the tendency of smaller non-polar surface MRA during stadials with the notable exception of HS1 in LOVECLIM showing a more complex (rather opposite) dynamic (Figure 3g)." We suggest to revise the sentence by splitting it in 2 parts: "Only when we calculate differences from simulations with and without abrupt AMOC shutdown during stadials for the same model do we find the tendency of smaller non-polar surface MRA during stadials (Figure 3g). However, a notable exception here is HS1 in LOVECLIM which shows a more complex (rather opposite) dynamic."

256: "for the same model do we find"?

AR: Already implemented in suggested change in the previous reply.

262: The manuscript likely needs some explanation of why / how there was a scaling of the eddy diffusivity to the AMOC proxy. This is unclear to me and likely other readers.

AR: We will add details here. In the sentence " Interestingly, scaling the eddy diffusivity to our AMOC proxy curve (SST record by Davtian and Bard, 2023) leads to a similar ΔMRA pattern in the outcrop-diffusion box model as in the other models when they are averaged in the low-latitude regime (Figure 3g)." we will correct "outcrop-diffusion box model" into "box-diffusion model" and we will add the following: "The linear scaling of the eddy diffusivity assumes that its value is 4000 m$^2$/yr (i.e. the modern value) throughout the Holocene and is reduced to 500 m$^2$/yr during the coldest interval of HS4 at ca. 40 kyr BP in the SST record (Fig. 1d). Instantaneous steady state MRA values are then calculated with the analytical equation 4 derived by Bard (1988)."

285: "dynamical behaviour" is not clear text. Does this refer to differences in model physics?

AR: This comment refers to "In addition to this common pattern, the models exhibit dynamical behaviour that is responsible for an increase of MRA in the northern North Atlantic, with varying intensity and spatial extent:" In models of the biogeochemical (BGC) cycle any differences between models are typically the combined effect of different model physics and different BGC. So, without further analysis it is not directly clear if physics (e.g.

ocean overturning) or BGC (e.g. air-sea 14C gas exchange) is responsible for the differences. For clarity we will revise the sentence into: "In addition to this common pattern, the models exhibit different dynamical behaviour (e.g. a different ocean overturning or a different air-sea gas exchange of $^{14}$C) that is responsible for an increase of MRA in the northern North Atlantic, with varying intensity and spatial extent"

285: Notably, there is no LSG model comparison here—maybe the manuscript should quickly state why?

AR: LSG is not included in this surface MRA anomaly for HS1-LGM, because the transient LSG simulations do not contain abrupt AMOC changes during Greenland stadials. This detail will be added here.

362: "…which model might be the.." (?)

AR: The comment refers to "The model-intercomparison shows where different models have strength and weaknesses, but leaves still open, which might be the best approach in the next iteration of IntCal, e.g. calculating multi-model means or relying on individual models." and suggest to add "model" after "which". We see that improvements might be a good idea, but suggest to change the sentence into: "The model-intercomparison shows where different models have strengths and weaknesses, but does not yet determine what the best approach in the next iteration of IntCal might be, e.g. calculating multi-model means or relying on individual models."
We furthermore will add the following sentence on the overall MIP: "While there are indeed important differences between the models that need to be understood, the overall level of agreement is fairly remarkable. This lends substantial support to the MarineCal endeavour and should strengthen community faith in the reliability of marine radiocarbon calibration curves (at least at low-latitudes in the open oceans)."

Figure 2: I like the layout of the panels. However, the caption confused me at first. I might replace "for runs with abrupt AMOC changes" with "for runs with modeled abrupt AMOC changes". When I first read this caption (before I read the paper), it almost sounded like the AMOC changes developed on their own!

AR: We will revise the caption of Figure 2 into "for runs with forced abrupt AMOC changes".